# CPDD: Generalized Compressed Representation for Multivariate Long-term Time Series Generation

## Abstract

The generation of time series has increasingly wide applications in many fields, such as electricity and transportation. Generating realistic multivariate long time series is a crucial step towards making time series generation models practical, with the challenge being the balance between long-term dependencies and short-term feature learning. Towards this end, we propose a novel time series generation model named Compressed Patch Denoising Diffusion-model (CPDD). Concretely, CPDD first employs the Time-series Patch Compressed (TPC) module based on the patch mode decomposition method to obtain the latent encoding of multi-scale feature fusion. Subsequently, it utilizes a diffusion-based model to learn the latent distribution and decode the resulting samples, thereby achieving high-quality multivariate long-time series generation. Through extensive experiments, results show that CPDD achieves state-of-the-art performance in the generation task of multivariate long-time series. Furthermore, TPC also exhibits remarkable efficiency in terms of robustness and generalization in time series reconstruction.

## 1 Introduction

The analysis of long-term time series data is of paramount importance in many real-world applications (Zhou et al. (2022); Wu et al. (2021); Zhou et al. (2021); Wang et al. (2023)). For instance, in power load forecasting, longer historical data spanning months or even years is crucial for capturing seasonal patterns and long-term trends, enabling more accurate predictions of future energy demand. However, the increasing complexity and volume of time series data, particularly in scenarios involving long sequences and multiple variables, present significant challenges for analysis and modeling. These challenges are further exacerbated by limited access to high-quality, diverse datasets, especially when data sharing is restricted due to privacy concerns or proprietary reasons.

Time series generation has emerged as a promising solution to address these issues. By artificially creating realistic and diverse time-series data, researchers and practitioners can overcome data scarcity and privacy constraints. Generating long time series presents its own set of difficulties, including maintaining temporal consistency, capturing complex long-term dependencies and short-term features, and ensuring the generated data accurately reflects the statistical properties of real-world time series.

To address the challenges of time series generation, various deep learning approaches have been proposed in recent years. These approaches aim to create synthetic time series data that closely resemble real-world data in terms of statistical properties and temporal dynamics. Some notable approaches include TimeGAN (Yoon et al. (2019)), TimeVAE (Abhyuday Desai & Beaver (2021)), and Diffusion-TS (Yuan & Qiao (2024)). TimeGAN leverages the Generative Adversarial Networks (GANs) framework (Goodfellow et al. (2014)) and introduces the supervised loss in training to improve the generation quality. TimeVAE utilizes variational auto-encoders to capture and reproduce the underlying distribution of time series data using the trend-season decomposition technique. More recently, diffusion-based models such as Diffusion-TS have shown promising results in generating high-quality time series data. However, these existing approaches face several challenges when it comes to generating multivariate long-term time series: 1) Cumulative errors: Recurrent

Figure 1: The patches of time series are represented as latent vectors, where the proximity of the vectors indicates the similarity between the patches. The time series will be reconstructed by decoding the combined latent vectors.

Neural Networks(RNN-based) methods exhibit limited performance in capturing long-range dependencies due to cumulative errors. 2) High computational demands: Transformer-based approaches lead to quadratic memory complexity for sequence length, requiring substantial computational resources in long-term time series generation. 3) Difficulty in capturing both long-term dependencies and short-term features: As the length of time series and the number of variables increase, the short-term features in time series will become increasingly complex due to additional seasonal changes, cyclical changes, sudden events, and other short-term fluctuations. Simultaneously, this heightened complexity may lead to long-term dependencies becoming more prominent and intricate. These challenges pose difficulties for current approaches in producing high-quality multivariate long-term time series that accurately reflect the complex temporal dynamics found in real-world scenarios.

One viable solution to the aforementioned issue is to employ a framework akin to the latent layer diffusion model (Rombach et al. (2022)), necessitating techniques capable of effectively compressing and characterizing multivariate long-term time series data. In this paper, we introduce a novel multivariate long-term time series generative model named Compressed Patch Denoising Diffusion-model (CPDD), which utilizes the mode functions decomposition technique to achieve cross-scale features fusion for obtaining the integrated representation of long-term dependence and short-term features. Concretely, We employ a Time-series Patch Compression (TPC) module of CPDD to decompose the patches into mode functions (Dragomiretskiy & Zosso (2014)), enabling a consistent characterization of both long-term dependencies and short-term features through these mode functions. In contrast to conventional time series representation approaches (Chowdhury et al. (2022); Yue et al. (2022); Zheng & Zhang (2023)), the TPC module learns a versatile combination of mode functions from patches to accommodate various patterns and achieve generalized compressed representation of time series data. To learn the latent distribution of the TPC outputs, CPDD introduces a trend-seasonal decomposition diffusion-based generative model (Ho et al. (2020)) which utilizes a novel Convolutional Neural Networks (CNN-based) Backbone named Depthwise separable Star Convolution (DSConv) block to capture complex nonlinear features and Transformer blocks to learn the trend in time series.

The major contributions of this paper are as follows:

- We propose CPDD, a novel compression-based denoising diffusion generative model designed for multivariate long-term time series. CPDD leverages a patch compression method to capture complex long-term and short-term dependencies more effectively, ultimately leading to the high-quality generation of multivariate long-term time series.

- We introduce the TPC module, which guides the model to learn the general mode functions of the patches and then fits it by mode functions combination to obtain the generalized time-compressed representation model, based on the patch mode decomposition technique. The TPC module can achieve robust reconstruction under high noise levels on some unseen time series data.

- We propose a novel CNN-based backbone network named DSConv block apply to adaptively learn the patch mode functions, which employs an element-wise multiplication operation akin to the kernel trick, integrating the Depthwise separable Convolution (DWConv) (Howard (2017)) technique with the ConvFFN structure (Luo & Wang (2024)) to dynamically learn the general mode functions.

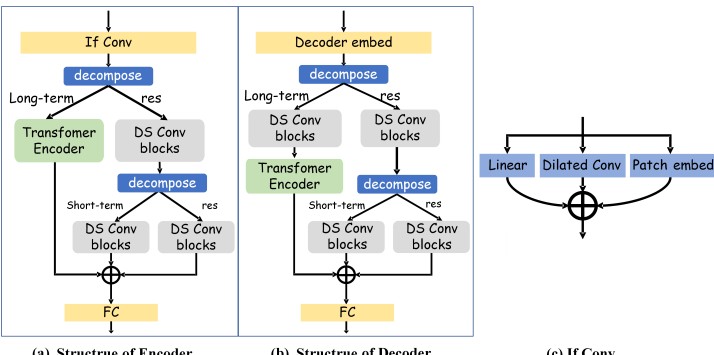

(a) Structrue of Encoder      (b) Structure of Decoder      (c) If Conv

Figure 2: The design of the TPC module. Integrated Feature Convolution (If Conv) is the embedding layer of the TPC module. $\bigoplus$ define as Element-wise addition. The decompose module employs a Moving average decomposing technique.

- We utilize various time series datasets to assess CPDD, showcasing its state-of-the-art performance in generating multivariate long-term time series. Furthermore, we evaluate the TPC module using various unseen datasets with noise levels of $30\%, 50\%$ in the task of time series reconstruction. Experimental results demonstrate remarkable robustness and generalization of the TPC module.

## 2 PROBLEM STATEMENT

Given a multivariate time series signal $\mathbf{X}_{1:\tau} \in \mathbb{R}^{\tau \times D}$, we split it into $N$ patches denoted as $\mathbf{X}^1, \mathbf{X}^2, \cdots, \mathbf{X}^N$, where each $\mathbf{X}^n \in \mathbb{R}^{\frac{\tau}{N} \times D}$ represents a segment of the original signal. Our goal is to decompose each patch into $K$ adaptive modes:

$$\mathbf{X}^n = \sum_{k=1}^{K} u_k + res_n,$$
$$u_k(t) = A_k \cdot \phi_k,$$
(1)

where $res_n \in \mathbb{R}^{\frac{\tau}{N} \times D}$ denotes irregular noise, $u_k \in \mathbb{R}^{\frac{\tau}{N} \times D}$ is the $k$-th adaptive mode function of the $n$-th patch, $A_k \in \mathbb{R}^{\frac{\tau}{N} \times D}$ is the amplitude function, and $\phi_k \in \mathbb{R}^{\frac{\tau}{N} \times D}$ is the basis function.

To maximize the fit of each patch, dense spaces of $u_k$ are necessary. Our objective is for the model to learn a dense representation space and extract a fitting combination of modes to reconstruct the original patch, as depicted in Figure 1. This approach enables us to capture complex temporal patterns and interactions within the data.

Following the model representation, we acquire the latent distribution of the original data. By employing the diffusion method to model this latent distribution, we can accomplish time series generation.

## 3 CPDD: A COMPRESSION-BASED DIFFUSION MODEL FOR TIME SERIES GRENERATION

A widely adopted approach for handling multivariate long-term time series is to first compress its high-dimensional structure, followed by leveraging a generative model to capture its latent distribution. Existing approaches for achieving effective compressed representations include TCN-AE, VAE, and MAE (Wang et al. (2023); C. Zhang & Wu (2022); Zheng & Zhang (2023); Yue et al. (2022); Nie et al. (2023); Zerveas et al. (2021)). However, TCN-AE is vulnerable to overfitting due to its limited regularization, while VAE tends to generate overly smooth reconstructions owing to its inherent KL-divergence regularization. Additionally, MAE, which utilizes the Transformer structure, may result in high-dimensional encodings, making it unsuitable for compact representations.

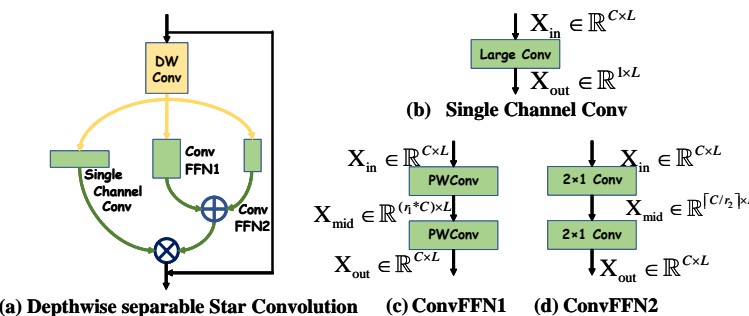

Figure 3: The design of Depthwise separable Star Convolution block. $L$ and $C$ are sizes of temporal and feature dimensions. DWConv and PWConv are short for depth-wise and point-wise convolution (Howard (2017)). $r_1, r_2 \in \mathbb{Z}_{>0}$, $\bigoplus$ define as Element-wise addition, $\bigotimes$ define as Element-wise multiplication. The output of the Single Channel Convolution will be broadcasted to match the dimension of the ConvFNN1 output.

Consequently, we aim to design a time series compression model that achieves a dense latent space and maintains robustness in generalization.

### 3.1 CPDD Method

CPDD is composed of the TPC module and a diffusion-based generative model, designed for efficient time series compression and reconstruction. The TPC module employs an encoder-decoder framework and integrates hierarchical components for long-term, short-term, and residual processing, which facilitate the learning of multi-scale temporal features and patch mode functions. The diffusion generative model, built upon the Transformer architecture, aims to capture the distribution of compressed representations more effectively.

### 3.2 Time-series Patch Compression module

The TPC module is composed of the DSConv block and the Transformer block components, as shown in Figure 2 and Figure 3. The DSConv block is employed to handle short-term and residual features due to its advanced complex nonlinear modeling capabilities, while the Transformer block is used to deal with long-term dependencies.

**Multi-scale Feature Extraction.** Several studies, such as ModernTCN (Luo & Wang (2024)), have demonstrated that CNNs can significantly improve temporal feature extraction in time series modeling by expanding the receptive field. To construct a comprehensive initial representation of time series data, we combine linear layers, dilated convolutions, and patch embedding layers (C. Zhang & Wu (2022); Zheng & Zhang (2023)) to form the embedding module, referred to as Integrated Feature Convolution (If Conv). This module is specifically designed to capture multi-scale features by effectively converting patches into representative tokens.

**Adaptive mode Function Learning.** To achieve adaptive learning of mode functions, we aim to design a model that learns a latent space where each point corresponds to a unique mode function, with smoothly distributed representations for both long-term and short-term functions. This enables capturing diverse morphological characteristics across different time scales. We propose the DSConv block, which incorporates structural regularization to support this adaptive learning. In the following subsections, we first describe the components of DSConv and then provide a detailed formal proof of the effectiveness of its structural regularization.

As illustrated in Figure 3, the DSConv block integrates a modified StarNet structure (Ma et al. (2024)) with three core components: DWConv, Single Channel Convolution, and two ConvFFN modules. Concretely, the DWConv is responsible for learning temporal multi-scale information. The Single Channel Conv aggregates information across channels to capture global dependencies. The ConvFFN modules, analogous to the Feed-Forward Networks (FFN) in Transformers,

utilize convolution operations for feature transformation. ConvFFN1 employs an inverted bottle-neck structure with two pointwise convolutions, which enables it to concentrate on a narrow field of view, facilitating the acquisition of general short-term feature information. ConvFFN2 utilizes an information-constrained bottleneck structure to emphasize the interdependence among neighboring tokens. Utilizing the star operation of StarNet, the DSConv block effectively fuses scale features, general mode features, and local dependencies into a unified representation. This operation enables multi-scale feature interaction and hierarchical representation learning, while preserving computational efficiency and enhancing the robustness of feature extraction across different temporal scales.

From the mathematical analysis, the star operation can be rewritten as follows:

$$(\mathbf{W_1}\mathbf{X} + b_1) \cdot (\mathbf{W_2}\mathbf{X} + b_2)$$

$$= \left(\sum_{g=1}^{G}\sum_{c=1}^{C} w_1^{g,c} x_{t+g-1,c} + b_1\right) \cdot \left(\sum_{g=1}^{G}\sum_{c=1}^{C} w_2^{g,c} x_{t+g-1,c} + b_2\right)$$

$$= \sum_{g_1=1}^{G}\sum_{c_1=1}^{C}\sum_{g_2=1}^{G}\sum_{c_2=1}^{C} \alpha_{g_1,c_1,g_2,c_2} x_{t+g_1-1,c_1} x_{t+g_2-1,c_2} \tag{2}$$

$$+ \sum_{g=1}^{G}\sum_{c=1}^{C} \beta_{g,c} x_{t+g-1,c} + \gamma,$$

$$\alpha_{g_1,c_1,g_2,c_2} = w_1^{g_1,c_1} w_2^{g_2,c_2}, \beta_{g,c} = w_1^{g,c} b_2 + w_2^{g,c} b_1, \gamma = b_1 b_2,$$

where $G \in \mathbb{N}^+$ represents the kernel size of the convolution and $g$ is the index. $C \in \mathbb{N}^+$ denotes the number of input channels. $x_{t,c} \in \mathbb{R}^1$ is the value at position $t$ of channel $c$ in the input sequence $\mathbf{X}$. $w_1^{g,c}, w_2^{g,c} \in \mathbb{R}^1$ are elements of the convolution kernels $\mathbf{W}_1$ and $\mathbf{W}_2$, respectively. $b_1, b_2 \in \mathbb{R}^1$ are scalar bias terms. $\alpha_{g_1,c_1,g_2,c_2}, \beta_{g,c}, \gamma \in \mathbb{R}^1$ are the transformed coefficients that encapsulate information from the original convolution kernels and biases. By expressing the operation in this form, we can observe how the input dimensions interact quadratically and intuitively, and we can find that DSConv is more complex than the general additive convolution operation. Let's consider the role of the DSConv block at a more abstract level. The $l$-th layer of DSConv blocks can be concisely described as follows:

$$\mathbf{h}^{(l+1)} = (\mathbf{W}_1^l \mathbf{h}^{(l)}) \otimes (\mathbf{W}_2^l \mathbf{h}^{(l)}),$$
$$= (\mathbf{A}^l \otimes \phi^l), \tag{3}$$

where $\mathbf{h}^{(l)} = \begin{bmatrix} \mathbf{X}^l \\ 1 \end{bmatrix}$ is the hidden representation at layer $l$, $\mathbf{W}_1^l = \begin{bmatrix} \mathbf{W}_1^l \\ \mathbf{b}_1^l \end{bmatrix}$ and $\mathbf{W}_2^l = \begin{bmatrix} \mathbf{W}_2^l \\ \mathbf{b}_2^l \end{bmatrix}$ are learnable weight matrices, $\otimes$ denotes element-wise multiplication. The next layer can be described as:

$$\mathbf{h}^{(l+2)} = (\mathbf{W}_1^{l+1}\mathbf{h}^{(l+1)}) \otimes (\mathbf{W}_2^{l+1}\mathbf{h}^{(l+1)}),$$
$$= (\mathbf{W}_1^{l+1}((\mathbf{W}_1^l\mathbf{h}^{(l)}) \otimes (\mathbf{W}_2^l\mathbf{h}^{(l)}))) \otimes (\mathbf{W}_2^{l+1}((\mathbf{W}_1^l\mathbf{h}^{(l)}) \otimes (\mathbf{W}_2^l\mathbf{h}^{(l)}))),$$
$$= (\mathbf{W}_1^{l+1}((\mathbf{A}^l \otimes \phi^l)) \otimes (\mathbf{W}_2^{l+1}((\mathbf{A}^l \otimes \phi^l)),$$
$$= (\mathbf{A}^{l+1} \otimes \phi^{l+1}). \tag{4}$$

Through iterative layer-by-layer processes, we can formalize the final equation as follows:

$$\mathbf{h}^{(L)} = (\mathbf{A}^{L-1} \otimes \phi^{L-1}). \tag{5}$$

Based on the iterative Equations (3)-(5) provided above, DSConv blocks can be approximated as the product of the magnitude function and the basis function, which the channel dimension is equivalent to $2^L$-exponent of $C/2^{0.5}$ (Ma et al. (2024)).

**Training Objectives.** We employ the L2 loss as the reconstruction loss to train the TPC model. The loss function is defined as:

$$\mathcal{L}_{recon} = ||\mathbf{X} - \hat{\mathbf{X}}||^2. \tag{6}$$

## 3.3 DIFFUSION GENERATIVE MODEL

As shown in Figure 4, We utilize the time-series trend-seasonal decomposition diffusion generative model as the generative module of CPDD, which is inspired by the Diffusion-TS model (Yuan & Qiao (2024)). However, the distinction lies in its utilization of the DSconv module, which replaces the functionality of the seasonal module that operates in parallel with the FFN. This allows for parallel computation with the entire Transformer decoder, thereby more effectively capturing complex temporal dynamics.

The diffusion model consists of two main components: a forward process and a reverse process. In the forward process, starting from an initial sample $z_0 \sim q(z)$, Gaussian noise is gradually added over $T$ steps, transforming it into a noise distribution $z_T \sim \mathcal{N}(0, I)$. This process is defined as:

$$q(z_t|z_{t-1}) = \mathcal{N}(z_t; \sqrt{1-\beta_t}z_{t-1}, \beta_t I), \tag{7}$$

where $\beta_t$ determines the noise level at each diffusion step $t$. The reverse process, modeled by $p_\theta$, aims to reconstruct the original sample by estimating:

$$p_\theta(z_{t-1}|z_t) = \mathcal{N}(z_{t-1}; \mu_\theta(z_t, t), \Sigma_\theta(z_t, t)). \tag{8}$$

The loss function minimizes the gap between the true posterior mean $\mu(z_t, z_0)$ and the predicted mean $\mu_\theta(z_t, t)$:

$$\mathcal{L}(\theta) = \sum_{t=1}^{T} \mathbb{E}_{q(z_t|z_0)} \left[ \|\mu(z_t, z_0) - \mu_\theta(z_t, t)\|^2 \right]. \tag{9}$$

**Trend block.** The trend component in this model aims to capture the underlying smooth trajectory of the data, representing gradual, long-term changes. We use the polynomial-based trend architecture (Boris N. Oreshkin & Bengio (2020)) to model the trend component $V_{tr}^n$. The formulation is as follows:

$$V_{tr}^t = \sum_{l=1}^{L} (\mathbf{F} \cdot Linear(o_{tr}^{t,l}) + \mu_{tr}^{t,l}), \tag{10}$$

where, $\mu_{tr}^{t,l}$ represents the mean value of the output from the $l$-th decoder block, $\mathbf{F} = [1, f, \cdots, f^p]$ denotes the slow-varying poly space, $f = [0, 1, 2, \cdots, n-2, n-1]^N/n]$, and $o_{tr}^{t,l}$ is the input of the $l$-th decoder block. The polynomial degree $p$ is intentionally kept low (typically $p \leq 3$) to ensure the model captures only gradual, low-frequency variations in the data.

**Seasonal & Residual block.** To fit components other than the trend, we try to utilize the previously mentioned DSConv block to learn the intricate non-linear features. After compression, both seasonal and residual irregular features become more pronounced and distinguishable within the dense latent distribution, making them easier to learn and capture. We view the model capturing Seasonal & Residual component process as the mode function learning process with the patch size 1 as follows:

$$\begin{aligned} u_n^{t,l} &= DSConv_l(o_{ds,n}^{t,l}) \\ &= A_n^{t,l} \cdot \phi_n^{t,l}, \end{aligned} \tag{11}$$

where $u_n^{t,l}$ is mode function of the $n$-th patch in the $l$-th decoder block, $o_{ds,n}^{t,l}$ denotes the $n$-th patch of the the $l$-th decoder block input. Combining the trends with each patch eventually, we obtain the original latent variables as follows:

$$\hat{z}(z_t, t, \theta) = V_{tr}^t + \left[ \sum_{l=1}^{L} u_1^{t,l}, \sum_{l=1}^{L} u_2^{t,l}, \cdots, \sum_{l=1}^{L} u_N^{t,l} \right] + res. \tag{12}$$

**Training Objectives.** We train the generative model to directly predict and estimate the latent variable $\hat{z}_0(z_t, t, \theta)$ of the original time series. The reverse process of our diffusion model is approximated using the Equation (13):

$$z_{t-1} = \frac{\sqrt{\bar{\alpha}_{t-1}}\beta_t}{1-\bar{\alpha}_t}\hat{z}_0(z_t, t, \theta) + \frac{\sqrt{\alpha_t}(1-\bar{\alpha}_{t-1})}{1-\alpha_t}z_t + \frac{1-\bar{\alpha}_{t-1}}{1-\bar{\alpha}_t}\beta_t\varepsilon_t, \tag{13}$$

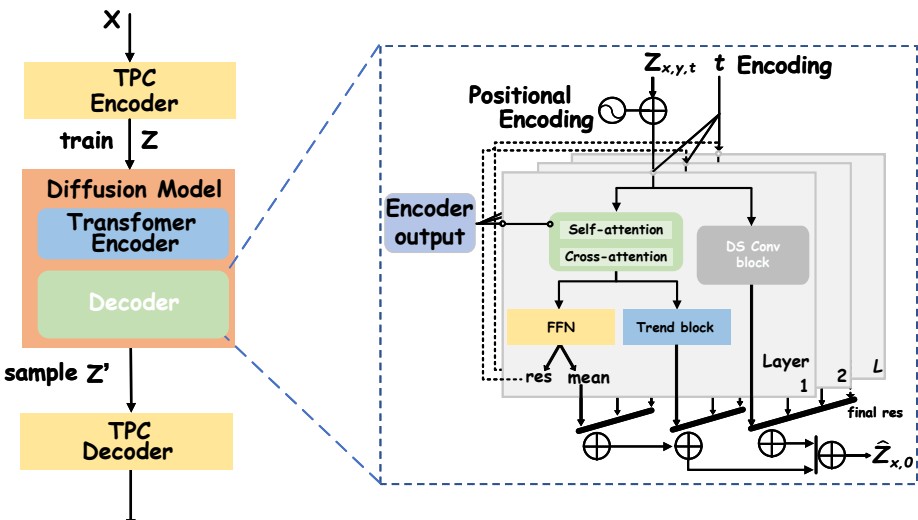

Figure 4: The detailed construction of the decoder in the diffusion-based generative model. $Z$: Latent representation obtained from the TPC Encoder during training. $Z_t$: Time-step latent variable used in the intermediate stages of the diffusion process. $Z'$: Latent sample generated by the diffusion model for reconstruction. $\hat{Z}_0$: Final denoised latent representation after the reverse diffusion process.

where $\varepsilon_t \sim \mathcal{N}(0,1)$, $\alpha_t = 1 - \beta_t$, and $\bar{\alpha}_t = \prod_{s=1}^{t} \alpha_s$. We employ a reweighting strategy for the loss function,

$$\mathcal{L}_{simple} = \mathbb{E}_{t,z_0}\left[ w_t|z_0 - \hat{z}_0(z_t, t, \theta)|^2 \right], \quad w_t = \frac{\lambda \alpha_t(1 - \bar{\alpha}_t)}{\beta_t^2}, \tag{14}$$

where $\lambda$ is a constant, typically set to 0.01. Due to the reconstruction error present in the TPC module, it is essential to incorporate the Mean Squared Error (MSE) between the original time series $\mathbf{X}$ and the decoded prediction time series $\hat{\mathbf{X}}$ as an extra loss term to aid in the convergence of the generative module.

$$\mathcal{L}_{finetun} = ||\mathbf{X} - \hat{\mathbf{X}}||^2. \tag{15}$$

To encourage the model to capture the temporal dependencies present in the data, we introduce an autocorrelation loss term. This loss penalizes the difference between the autocorrelation function of the generated sequences and the target autocorrelation structure. Formally, we define the autocorrelation loss as:

$$\mathcal{L}_{ACF} = \sum_{m=1}^{M} w(m)|\hat{r}(m) - r(m)|^p, \tag{16}$$

where $\hat{r}(m)$ is the sample autocorrelation function of the model-generated sequence at lag $m$, $r(m)$ is the target autocorrelation function, $w(m)$ is a weighting function that allows us to emphasize certain lags and $p$ is the norm parameter (typically 1 or 2). The summation is carried out up to a maximum lag $m$, chosen based on the relevant time scales in our data. The total loss function combines multiple objectives:

$$\mathcal{L}_{total} = \mathcal{L}_{simple} + \lambda_{ACF}\mathcal{L}_{ACF} + \lambda_{finetun}\mathcal{L}_{finetun}. \tag{17}$$

# 4 EMPIRICAL EVALUATION

In this section, we present a comprehensive evaluation of the CPDD model. We compare CPDD with the baseline generation models: Diffusion-TS, TimeGAN, and TimeVAE. Our experiments are designed to validate the effectiveness of CPDD across various scenarios, including quality of generation, component contributions, robustness, and generalization of reconstruction.

## 4.1 EXPERIMENTAL SETUP

**Dataset.** We evaluate our method using diverse datasets representing various aspects of time series analysis: 1) Sines Dataset has 10 features where each feature is created with different frequencies and phases independently. 2) Electricity Dataset contains hourly electricity consumption data from 370 clients in New South Wales, Australia, spanning from 1996 to 1998, and is commonly used for time series forecasting and detecting anomalies in power usage patterns. We limit the training data for the electricity dataset to only the first 7 clients. 3) ETTh Dataset is the electricity Transformer monitoring data comprised of multivariate time series data with 7 features recorded at 15-minute intervals spanning from July 2016 to July 2018, encompassing load and oil temperature metrics. 4) Energy Dataset is from the UCI appliance energy prediction suite, comprising 28 variables for energy consumption pattern analysis.

**Metrics.** We employ the **Discriminative Score** to evaluate the realism of synthetic data by distinguishing it from real samples, and the **Predictive Score** to measure its predictive value by training on synthetic data and testing on real data (Yoon et al. (2019)).

## 4.2 GENERATION EXPERIMENTS

In this subsection, we evaluate the performance of CPDD in time series generation tasks, comparing it with other baselines across various datasets.

Table 1: Results on Multiple Time-Series Datasets (Bold indicates best performance).

| Metric | Methods | Sines | Electricity | ETTh | Energy |
|---|---|---|---|---|---|
| Discriminative Score (Lower the Better) | CPDD (ours) | **0.272±0.175** | **0.355±0.112** | **0.352±0.082** | **0.488±0.004** |
| | Diffusion-TS ( Yuan & Qiao (2024)) | 0.495±0.003 | 0.497±0.002 | 0.492±0.007 | 0.499±0.001 |
| | TimeGAN ( Yoon et al. (2019)) | 0.499±0.001 | 0.487±0.012 | 0.488±0.011 | 0.499±0.001 |
| | TimeVAE ( Abhyuday Desai & Beaver (2021)) | 0.483±0.016 | 0.360±0.082 | 0.360±0.081 | 0.499±0.001 |
| Predictive Score (Lower the Better) | CPDD (ours) | 0.326±0.001 | 0.625±0.028 | **0.751±0.021** | 0.976±0.008 |
| | Diffusion-TS ( Yuan & Qiao (2024)) | **0.311±0.002** | 0.803±0.072 | 1.287±0.023 | 0.995±0.001 |
| | TimeGAN ( Yoon et al. (2019)) | 0.352±0.004 | 0.708±0.037 | 0.922±0.038 | **0.926±0.002** |
| | TimeVAE ( Abhyuday Desai & Beaver (2021)) | 0.348±0.018 | **0.597±0.015** | 0.762±0.023 | 0.927±0.001 |
| | Original | 0.311±0.001 | 0.548±0.001 | 0.784±0.008 | 0.825±0.009 |

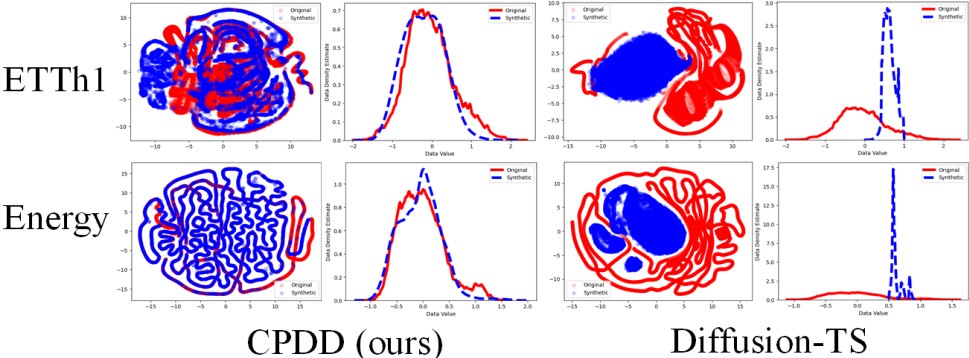

ETTh1

Energy

CPDD (ours)         Diffusion-TS

Figure 5: Visualizations of the time series synthesized by Diffusion-TS and CPDD (Blue for synthetic data, red for raw data).

**Results and Analysis.** Table 1 displays the comparative results of generating 1024-length time series data using CPDD and baseline models. The test results indicate that CPDD outperforms other models on the majority of the datasets. The performance gap is more pronounced in the Discriminative score at all datasets, highlighting the efficacy of CPDD in capturing temporal dynamics. To assess the similarity between synthetic and real-time series data distributions, we employ two complementary visualization approaches. We first use t-SNE (van der Maaten & Hinton (2008)) to project original and synthetic data into a lower-dimensional space for visual comparison of clustering patterns. Then, we apply Kernel Density Estimation (KDE) to compare their probability density functions, highlighting distributional similarity. From Figure 5, the visualization results show that the time series synthesized by CPDD has better distribution coincidence than the other baselines, demonstrating the effectiveness of our method in modeling long-term and short-term time series. The comprehensive evaluation demonstrates the superior performance of CPDD in time series generation across various datasets.

## 4.3 ABLATION STUDIE

We compare the full versioned CPDD with its two variants to evaluate the effectiveness of our method: (1) w/o DSConv, i.e. CPDD replaces the DSConv block with FFT-based block in the diffusion-base generative model. (2) w/o Compress, i.e. CPDD without the TPC module. The details of the results are shown in Table 2.

Table 2: Ablation study for model architecture and options. (Bold indicates best performance).

| Metric | Method | Sines | Electricity | ETTh | Energy |
|---|---|---|---|---|---|
| Discriminative Score (Lower the Better) | CPDD | **0.272±0.175** | **0.355±0.112** | **0.352±0.082** | **0.488±0.004** |
| | w/o DSConv | 0.352±0.133 | 0.496±0.003 | 0.497±0.001 | 0.495±0.004 |
| | w/o TPC | 0.494±0.003 | 0.499±0.001 | 0.495±0.004 | 0.498±0.002 |
| Predictive Score (Lower the Better) | CPDD | 0.326±0.001 | **0.625±0.028** | **0.751±0.021** | 0.972±0.003 |
| | w/o DSConv | **0.318±0.002** | 0.698±0.015 | 0.763±0.073 | 0.972±0.003 |
| | w/o TPC | 0.333±0.001 | 0.786±0.012 | 1.039±0.083 | **0.953±0.011** |
| | Original | 0.311±0.001 | 0.548±0.001 | 0.784±0.008 | 0.825±0.009 |

Results indicate that both the DSConv and the TPC module design contribute significantly to the performance of the model. We find that removing either the TPC module or the DSConv component results in a notable drop in performance, demonstrating their crucial roles. The TPC module efficiently compresses both long- and short-term features, while the DSConv enhances multi-scale feature interactions, together ensuring robust and high-quality time series generation.

## 4.4 ROBUSTNESS ANALYSIS

In this part, we test the reconstruction robustness of the TPC module. The validation model is only trained by the training set of ETTh1, which means that all test sets of this experiment are unseen for the validation model. As the feature dimensions in some datasets vary, we randomly select 7 dimensions for high-dimensional datasets and utilize cyclic padding to fill the gaps in datasets with less than 7 dimensions. We add varying levels of noise to the input data, focusing on the performance of the time series compression module to test the robustness of the model. From Table 3, the model demonstrates excellent reconstruction ability on the majority of unseen datasets, showcasing its strong generalization performance. Additionally, it maintains good reconstruction performance even at noise levels of 30% and 50%, highlighting the robustness of the model in reconstruction tasks.

Table 3: Reconstruction performance under different noise levels. (The test value means the MSE)

| Noise | ETTh1(baseline) | ETTh2 | ETTm1 | ETTm2 | Electricity | Exchange rate | Traffic | stock | Energy | Wheather |
|---|---|---|---|---|---|---|---|---|---|---|
| 0% | 0.0577 | 0.0337 | 0.0459 | 0.0198 | 0.0726 | 0.0541 | 0.182 | 0.0522 | 0.0668 | 0.0936 |
| 30% | 0.078 | 0.0592 | 0.0812 | 0.0469 | 0.0937 | 0.0764 | 0.210 | 0.0731 | 0.0948 | 0.113 |
| 50% | 0.116 | 0.105 | 0.145 | 0.0965 | 0.132 | 0.116 | 0.260 | 0.112 | 0.146 | 0.147 |

## 5 CONCLUSSION

In this work, we introduce a novel approach for realistic multivariate long-time series generation through a model named CPDD. CPDD effectively addresses the fundamental challenges of balancing long-term temporal dependencies and short-term feature representations by integrating the TPC module and a diffusion-based generative model. The TPC module leverages adaptive patch mode decomposition to capture and compress multi-scale temporal features, ensuring a robust and compact latent encoding. The diffusion-based framework then models the latent distribution and synthesizes high-quality long-time series data through an iterative denoising process. Extensive experiments validate that CPDD not only achieves state-of-the-art performance in generating complex multivariate long-term time series but also demonstrates remarkable generalization and robustness in various time series reconstruction tasks. In future work, we will focus on enhancing the TPC module and DSConv block to handle more complex temporal structures and explore their integration into hybrid architectures for broader applicability across diverse domains.

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

# A    APPENDIX

## A.1    COMPUTATION COMPLEXITY

Table 4 presents the results of the training and sampling times in comparison to Disffusion-TS and CPDD. Table 5 displays the corresponding hyperparameter settings for this experiment. Based on both the actual experimental results and the theoretical computational complexity analysis, it is evident that CPDD requires less time for training and sampling compared to Diffusion-TS. This efficiency is attributed to the effective Time-series Patch Compression Encoder in CPDD, which significantly reduces the length of input time series without adding excessive feature dimensions.

Table 4: Comparison of Training and Sampling Times for Diffusion-TS and CPDD

| Dataset | Model | Training Time(min) | Sample Time(min) |
|---|---|---|---|
| sines | Diffusion-TS | 146 | 65 |
| | CPDD | **63** | **30** |
| electricity | Diffusion-TS | 109 | 495 |
| | CPDD | **92** | **65** |
| Etth1 | Diffusion-TS | 181 | 111 |
| | CPDD | **74** | **42** |
| Energy | Diffusion-TS | 185 | 410 |
| | CPDD | **128** | **38** |

Table 5: Comparison of Key Parameters between Diffusion-TS and CPDD

| Parameter | Diffusion-TS | CPDD |
|---|---|---|
| Input Size | 1024 | 1024 |
| Channels | 7 | 7 |
| Dataset | ETTh1 | ETTh1 |
| Number of samples | 17,017 | 17,017 |
| Encoder Layers | 3 | 3 |
| Decoder Layers | 2 | 2 |
| $d_{\mathrm{model}}$ | 64 | 64 |
| TPC Encoder Layers | - | 1 |
| TPC Decoder Layers | - | 1 |
| TPC Encoder $d_{\mathrm{model}}$ | - | 256 |
| TPC Encoder Token Size | - | 64 |
| TPC Decoder $d_{\mathrm{model}}$ | - | 128 |
| TPC Decoder Token Size | - | 64 |
| Computational Complexity | $272 \times 5 \times 64^3$ | $(26 + 10) \times 64^3$ |

## A.2    COMPARATIVE EXPERIMENTS ON PATCH SIZE

In Table 6, the comparative experimental results of the 1024th generation of the ETTh1 dataset under various patch size conditions are presented, while Table 7 illustrates the corresponding hyperparameter settings.

Table 6: Discriminative and Predictive Scores for ETTh1 with Different Patch Sizes

| Patch Size | Discriminative Score↓ | Predictive Score↓ |
|---|---|---|
| 8 | 0.412 ± 0.037 | **0.696 ± 0.037** |
| 10 | 0.407 ± 0.042 | 0.732 ± 0.017 |
| 16(CPDD) | **0.352 ± 0.082** | 0.751 ± 0.021 |

The experimental results indicate that as the patch size decreases, there is a gradual increase in the Discriminative score while the Predictive score decreases gradually, which aligns with our theoretical assumption. Nevertheless, their performance remains superior to most baseline performances in Table 1.

Table 7: Parameter Configuration for Different Patch Sizes

| Parameter | | | |
|---|---|---|---|
| Patch size | 8 | 10 | 16 |
| Stride | 6 | 8 | 16 |
| Input Size $L$ | 1024 | 1024 | 1024 |
| Channels | 7 | 7 | 7 |
| Diffusion Model Encoder Layers | 1 | 3 | 3 |
| Diffusion Model Decoder Layers | 2 | 2 | 2 |
| Diffusion Model Token length | 128 | 103 | 64 |
| Diffusion Model Feature dimension $D_{tz}$ | 64 | 64 | 64 |
| DSConv of Diffusion Model Feature dimension $D_{sz}$ | 128 | 256 | 384 |
| TPC Encoder Layers | 1 | 1 | 1 |
| TPC Decoder Layers | 1 | 1 | 1 |
| TPC Encoder Feature dimension $D_{te}$ | 128 | 256 | 256 |
| TPC Encoder Token length | 171 | 128 | 64 |
| TPC Decoder Feature dimension $D_{td}$ | 64 | 128 | 128 |
| TPC Decoder Token length | 171 | 128 | 64 |
| TPC DSConv Feature dimension $D_s$ | 16 | 25 | 32 |
| TPC DSConv Token Size | 171 | 128 | 64 |

## A.3 SUPPLEMENT TO THE EVALUATION METRICS

We add the Context-FID Score (Jeha et al. (2022)) and Correlational Score (Liao et al. (2020)) eval-
uation metrics to the Table 8 and Table 9. Context-FID involves extracting features of generated
and real sequences using a pre-trained embedding model and calculating the distribution difference
in the embedding space to measure the overall quality of the generated data. The Correlational
method compares the autocorrelation and cross-correlation distributions of generated data with real
data to assess whether it preserves the statistical structure and dependency patterns of the time se-
ries. Overall, CPDD demonstrates excellent generation performance, excelling in both Context-FID
and Correlational metrics, while also maintaining a good balance between Context consistency and
multivariate time series correlation.

Table 8: Context-FID Scores for Different Models Across Datasets (Lower is Better).

| Model | Sines | Electricity | ETTh1 | Energy |
|---|---|---|---|---|
| CPDD | **5.687 ± 0.252** | 5.996 ± 0.294 | **3.238 ± 0.421** | **1.151 ± 1.072** |
| Diffusion-TS | 13.451 ± 0.492 | 65.204 ± 1.853 | 20.568 ± 2.973 | 67.630 ± 7.732 |
| TimeGAN | 59.031 ± 2.223 | 18.365 ± 0.613 | 10.381 ± 1.227 | 61.022 ± 1.893 |
| TimeVAE | 106.981 ± 0.722 | **4.804 ± 0.436** | 3.362 ± 0.432 | 19.862 ± 0.038 |

Table 9: Correlational Scores for Different Models Across Datasets (Lower is Better).

| Model | Sines | Electricity | ETTh1 | Energy |
|---|---|---|---|---|
| CPDD | 0.329 ± 0.005 | 0.198 ± 0.001 | 0.247 ± 0.003 | **1.912 ± 0.001** |
| Diffusion-TS | **0.194 ± 0.003** | 0.213 ± 0.001 | 0.430 ± 0.008 | 7.271 ± 0.007 |
| TimeGAN | 1.392 ± 0.003 | 0.726 ± 0.002 | 1.811 ± 0.001 | 15.581 ± 0.003 |
| TimeVAE | 4.263 ± 0.001 | **0.086 ± 0.001** | **0.155 ± 0.004** | 2.484 ± 0.004 |

## A.4 SUPPLEMENT TO ABLATION EXPERIMENTS

To better evaluate the contribution of each component of CPDD, we added two ablation experiments
in Table 10. Firstly, to assess the impact of patch embedding, we introduced tests with patch=1 in
the table for comparison with the original patch=16. Secondly, to evaluate the effectiveness of
the trend-seasonal decomposition method, we conducted individual tests for the trend and seasonal
components.

Table 10: Discriminative and Predictive Scores for ETTh1 and Energy Datasets.

| Dataset | Method | Discriminative Score↓ | Predictive Score↓ |
|---|---|---|---|
| ETTh1 | Patch size = 1 | 0.499 ± 0.001 | 0.751 ± 0.011 |
| | Patch size = 16 (CPDD) | **0.352 ± 0.082** | **0.751 ± 0.021** |
| | Trend-only | 0.499 ± 0.001 | 0.792 ± 0.012 |
| | Season-only | 0.499 ± 0.001 | 0.789 ± 0.014 |
| Energy | Patch size = 1 | 0.499 ± 0.001 | **0.966 ± 0.001** |
| | Patch size = 16 (CPDD) | **0.488 ± 0.004** | 0.972 ± 0.003 |
| | Trend-only | 0.499 ± 0.001 | 0.988 ± 0.002 |
| | Season-only | 0.499 ± 0.001 | 0.982 ± 0.004 |

The results shows that the CPDD method (Patch size = 16) achieves the best Discriminative Score across both datasets and competitive Predictive Score, demonstrating its effectiveness in capturing meaningful patterns.

