# OpenReview forum: "CPDD: Generalized Compressed Representation for Multivariate Long-term Time Series Generation"
_ICLR.cc/2025/Conference — Submitted to ICLR 2025_

### Official Review · Reviewer_fwsP · 2024-10-24

**Soundness:** 2
**Presentation:** 3
**Contribution:** 2
**Rating:** 5
**Confidence:** 4

**Summary:**

This paper analyses the challenges faced in the time series generation task, including the limited ability of the proposed approach to model long-term dependencies due to cumulative errors, the high computational complexity and time overhead due to the attention mechanism, and the inability to capture both long-term global dependencies and short-term local features. Inspired by the spatial distribution of latent variables modelled by LDM, in order to achieve a balance between long-term dependencies and short-run feature learning in time-series generation tasks, it proposes the Compressed Patch Denoising Diffusion-model (CPDD), where Time-series Patch Compressed (TPC) is designed based on the block pattern decomposition method to obtain multi-scale latent representations. And the diffusion model achieves high quality multivariate long time series generation after decoding by modelling the probability distribution of potential representations.

**Strengths:**

The main contribution of this paper is to propose a technique that can efficiently compress and represent multivariate long-term time series data by decomposing the patches into pattern functions through which long-term dependencies and short-term features are consistently represented. Specifically, the TPC module learns generic combinations of pattern functions from patches to accommodate various patterns and enables a generic compressed representation of time series data.

**Weaknesses:**

This paper is dedicated to present a technique that can efficiently compress and model multivariate long-term time series data, which has important real-world implications. The main concerns are as follows:

1.	As a model ‘designed for multivariate long-term time series’, the main innovative structures proposed by CPDD, DSConv and TPC, do not have a structure or design aimed at establishing cross-channel connectivity. We believe that a key question is whether the proposed single channel Convolution can establish connectivity across a large number of channels, e.g., the ECL dataset of the electricity Scenarios in the time series prediction contains 321 channels and the Traffic dataset contains 862 channels. Advances in multivariate prediction methods (iTransformer[1], SAMformer[2]) have shown that proper integration of channel management strategies in time series backbones is crucial for discovering univariate dynamics and cross-variate correlations.

2.	The lack of advanced baselines leads to the inability to validate the competitiveness of the proposed CPDD. Specifically, only three baselines based on Diffusion are shown in Table 1, and among them, TimeGAN and TimeVAE are published in 2021 and 2019, respectively. The introduction of a wider range of Baselines to compare the performance of the proposed models is expected to be complemented to fully validate the effectiveness of the proposed methods. The referenced baselines can be divided into 4 parts: 1) Models based on pre-trained LLM alignment to TS, e.g. TimeLLM[3]; 2) Pre-trained foundation models on unified time series datasets from multiple domains, e.g. Timer[4], UniTime[5]; 3) Proprietary models trained and tested on specific datasets, e.g. PatchTST[6]; 4) Recent Diffusion-based temporal probabilistic prediction models, e.g. Diffuison-TS[7], mr-Diff[8]. CCPD is expected to be compared with at least one competitive model in each prediction paradigm to demonstrate the soundness of the model design. In addition, we would like to introduce more benchmarks, such as ECL and Traffic datasets with a large number of channel counts, which we believe will help to validate the promising real-world applications of the proposed models.

3.	The design of the ablation experiments in this paper is deficient. In addition to DSConv and TPC, CPDD uses other strategies such as Patch Embed and Trend-seasonal Decomposition, yet the ablation experiments presented in Table 2 do not include these structural designs. This raises our concern about the validity of DSConv and TPC.

[1] Liu, Yong et al. “iTransformer: Inverted Transformers Are Effective for Time Series Forecasting.” ICLR 2024.
[2] Ilbert, Romain et al. “SAMformer: Unlocking the Potential of Transformers in Time Series Forecasting with Sharpness-Aware Minimization and Channel-Wise Attention.” ICML 2024.
[3] Jin, Ming et al. “Time-LLM: Time Series Forecasting by Reprogramming Large Language Models.” ICLR 2024.
[4] Liu, Yong et al. “Timer: Generative Pre-trained Transformers Are Large Time Series Models.” ICML 2024.
[5] Liu, Xu et al. “UniTime: A Language-Empowered Unified Model for Cross-Domain Time Series Forecasting.” Proceedings of the ACM on Web Conference 2024.
[6] Nie, Yuqi et al. “A Time Series is Worth 64 Words: Long-term Forecasting with Transformers.” ICLR 2023.
[7] Yuan, Xinyu and Yan Qiao. “Diffusion-TS: Interpretable Diffusion for General Time Series Generation.” ICLR 2024.
[8] Shen, Lifeng et al. “Multi-Resolution Diffusion Models for Time Series Forecasting.”  ICLR 2024.

**Questions:**

1.	The results of Baseline presented in Table 1 are inconsistent with the results presented in the original paper, e.g., the Discriminative Score of the Diffusion-TS model under the Sines dataset in Table 1 is 0.326, whereas it is reported as 0.006 in the original paper.In fact, the results of all Baselines in Table 1 are in significant differences. In addition, Table 1 only shows some of the metrics on the performance of time series generation, and the results of the proposed method on both Context-FID Score and Correlational Score are missing. In addition, traditional metrics for time series forecasting, such as MSE, MAE, CRPS, etc., are missing from Table 1, which results in the reader not getting a full picture of the potential limitations of CPDD.

2.	In Table 2, in the Predictive Score metric, the model with DSConv removed achieves better performance in the Sines dataset, and the model with TPC removed exhibits the best performance in the Energy dataset. The results of the ablation experiments are puzzling, which may shake the rationality of the structural design of DSConv and TPC.

---

> ### Author Response · Authors · 2024-11-22
> **Response to Reviewer fwsP**
>
> >**W  1**: As a model ‘designed for multivariate long-term time series’, the main innovative structures proposed by CPDD, DSConv and TPC, do not have a structure or design aimed at establishing cross-channel connectivity.We believe that a key question is whether the proposed single channel Convolution can establish connectivity across a large number of channels, e.g., the ECL dataset of the electricity Scenarios in the time series prediction contains 321 channels and the Traffic dataset contains 862 channels. Advances in multivariate prediction methods (iTransformer[1], SAMformer[2]) have shown that proper integration of channel management strategies in time series backbones is crucial for discovering univariate dynamics and cross-variate correlations.
>
> We apologize for the confusion caused by the unclear expression. CPDD has structures designed for managing cross-channels, specifically, the proposed DSConv in the paper inherits the point convolution structure of depthwise separable convolution, learning the relationships between different variables in patches through point convolution operations. Single-channel convolutions have convolution kernels with a large size (typically 4 times the depthwise convolution kernel), aiming to learn the relative scale relationships between different patch mode functions.
>
> As the reviewer mentioned, cross-channel correlations have an impact on time series tasks. However, the correlations between variables in time series datasets from different domains vary. For instance, in power data, the relationships between variables represent different load nodes, including spatial topology and the balance between grid load and generation. To systematically address the issue of generating cross-channel correlations in time series, it is crucial to consider the differences in time series from various domains.
>
> This challenge may require an additional paper to address. Generating long-time series (1024th and above) for power data and traffic data with hundreds of nodes is quite challenging.
>
> >**W  2 (Part A)**:  The lack of advanced baselines leads to the inability to validate the competitiveness of the proposed CPDD. Specifically, only three baselines based on Diffusion are shown in Table 1, and among them, TimeGAN and TimeVAE are published in 2021 and 2019, respectively. The introduction of a wider range of Baselines to compare the performance of the proposed models is expected to be complemented to fully validate the effectiveness of the proposed methods. The referenced baselines can be divided into 4 parts: 1) Models based on pre-trained LLM alignment to TS, e.g. TimeLLM[3]; 2) Pre-trained foundation models on unified time series datasets from multiple domains, e.g. Timer[4], UniTime[5]; 3) Proprietary models trained and tested on specific datasets, e.g. PatchTST[6]; 4) Recent Diffusion-based temporal probabilistic prediction models, e.g. Diffuison-TS[7], mr-Diff[8].
>
> Thanks to the reviewer for this valuable suggestion and the detailed categorization of baselines. One misunderstanding is that Timegan and TimeVAE are not diffusion models; only Diffusion-TS is a Diffusion model.
>
> We acknowledge the current lack of sufficient time series generation baselines for comparison. We will promptly test more time series generation baseline models and update them in the comments section or paper appendix.
>
> TimeLLM、Timer、UniTime and PatchTST and mr-Diff are outstanding contributions in the field of time series forecasting. However, they are not typically employed for direct time series generation.
>
> >**W  2 (Part B)**:  In addition, we would like to introduce more benchmarks, such as ECL and Traffic datasets with a large number of channel counts, which we believe will help to validate the promising real-world applications of the proposed models.
>
> Thanks for the reviewer's comment. We have measured the Electricity dataset and the Energy dataset in the Table 1 of the paper. As mentioned in weakness 1, generating ECL and Traffic datasets with a large number of channel counts poses significant challenges for both the CPDD model and current mainstream generation methods. This challenge is not only reflected in the model's generation capabilities but also in designing appropriate metric methods for these complex datasets.
>
> We have also dabbed in spatio-temporal predictive analytics in the electricity and transportation domains. We believe that the synthesized ECL and Traffic data should exhibit characteristics that adhere to real-world physical constraints and can be directly used for power and traffic analysis tasks. Addressing these challenges requires exploring the design of more specialized synthesis frameworks and evaluation methods, which would be better suited for publication in journals or conferences within the fields of power or traffic analysis.

---

> ### Author Response · Authors · 2024-11-22
> **Response to Reviewer fwsP**
>
> >**W 3**:The design of the ablation experiments in this paper is deficient. In addition to DSConv and TPC, CPDD uses other strategies such as Patch Embed and Trend-seasonal Decomposition, yet the ablation experiments presented in Table 2 do not include these structural designs. This raises our concern about the validity of DSConv and TPC.
>
> We apologize for not thoroughly testing the roles of fundamental components. We have added two ablation experiments. Firstly, to assess the impact of patch embedding, we introduced tests with patch=1 in the table for comparison with the original patch=16. Secondly, to evaluate the effectiveness of the trend-seasonal decomposition method, we conducted individual tests for the trend and seasonal components.
>
> | **Dataset** | **Method**             | **Discriminative Score↓** | **Predictive Score↓** |
> |--------------------------------|------------------------|--------------------------:|----------------------:|
> | **ETTh1**                      | Patch size = 1        |                      0.499±.001 |        **0.751±.011** |
> |                                | Patch size = 16 (CPDD)|        **0.352±.082**     |              **0.751±.021** |
> |                                | Trend-only            |                      0.499±.001 |              0.792±.012 |
> |                                | Season-only           |                      0.499±.001 |              0.789±.014 |
>
> | **Dataset** | **Method**             | **Discriminative Score↓** | **Predictive Score↓** |
> |--------------------------------|------------------------|--------------------------:|----------------------:|
> | **Energy**                     | Patch size = 1        |                      0.499±.001 |        **0.966±.001** |
> |                                | Patch size = 16 (CPDD)|        **0.488±.004**     |              0.972±.003 |
> |                                | Trend-only            |                      0.499±.001 |              0.988±.002 |
> |                                | Season-only           |                      0.499±.001 |              0.982±.004 |
>
> The table shows that the CPDD method (Patch size = 16) achieves the best Discriminative Score across both datasets and competitive Predictive Score, demonstrating its effectiveness in capturing meaningful patterns.
>
> >**Q 1 (Part A): The results of Baseline presented in Table 1 are inconsistent with the results presented in the original paper, e.g., the Discriminative Score of the Diffusion-TS model under the Sines dataset in Table 1 is 0.326, whereas it is reported as 0.006 in the original paper.In fact, the results of all Baselines in Table 1 are in significant differences.
>
> We apologize for the unclear expression that led to the reviewer's misunderstanding. In Diffusion-TS, the experimental setting involves comparing the generation of multi-variable time series with a length of 24th, while our experiment compares multi-variable time series with a length of 1024th.
>
> Additionally, we uniformly applied standard normalization in both Discriminative Score evaluation and Discriminative Score evaluation. While our approach primarily focuses on generating multivariate long-time series, we believe it is necessary to also test it on shorter time series to evaluate its functionality. We conducted tests on the sequence of lengths 64th in ETTh1, and the results are presented in the table below. (The baseline model results are cited from Diffusion-TS[1].)
>
> | Metric↓ | CPDD        | Diffusion-TS | TImegan      | Timevae      | Diffwave     | DiffTime     | Cot-GAN      |
> |---------------------|-------------|--------------|--------------|--------------|--------------|--------------|--------------|
> | Context-FID Score   | **0.495±.135** | 0.631±.058  | 1.130±.102   | 0.827±.146   | 1.543±.153   | 1.279±.083   | 3.008±.277   |
> | Correlational   Score | 0.183±.001  | 0.082±.005 | 0.483±.019   | **0.067±.006**  | 0.186±.008   | 0.094±.010   | 0.271±.007   |
> | Discriminative Score    | 0.112±.089  | **0.106±.048** | 0.227±.078   | 0.171±.142   | 0.254±.074   | 0.150±.003   | 0.296±.348   |
> | Predictive Score    | **0.102±.005** | 0.116±.000  | 0.132±.008   | 0.118±.004   | 0.133±.008   | 0.118±.004   | 0.135±.003   |
>
> Overall, our model demonstrates consistently strong performance in both long-time series and short-time series, showcasing its robustness across varying sequence lengths.

---

> ### Author Response · Authors · 2024-11-22
> **Response to Reviewer fwsP**
>
> >**Q 1 (Part B): In addition, traditional metrics for time series forecasting, such as MSE, MAE, CRPS, etc., are missing from Table 1, which results in the reader not getting a full picture of the potential limitations of CPDD.
>
> We thank the reviewer's suggestions. In generation tasks, multiple valid outputs are common. However, MAE and MSE calculations focus on point-to-point errors and do not consider the diversity or semantic similarity between the generated outcomes and the reference targets. We have added the Context-FID[2] and Correlational[3] evaluation metrics to the table below. Context-FID involves extracting features of generated and real sequences using a pre-trained embedding model and calculating the distribution difference in the embedding space to measure the overall quality of the generated data. The Correlational method compares the autocorrelation and cross-correlation distributions of generated data with real data to assess whether it preserves the statistical structure and dependency patterns of the time series.
>
> **Context-FID Scores↓**:
> | Model         | Sines          | Electricity     | ETTh1          | Energy          |
> |---------------|----------------|-----------------|----------------|-----------------:|
> | CPDD          | **5.687±.252** | 5.996±.294    | **3.238±.421** | **1.151±1.072** |
> | Diffusion-TS  | 13.451±.492  | 65.204±1.853    | 20.568±2.973   | 67.630±7.732    |
> | Timegan       | 59.031±2.223   | 18.365±.613     | 10.381±1.227   | 61.022±1.893  |
> | Timevae       | 106.981±.722   | **4.804±.436**  | 3.362±.432   | 19.862±.038     |
>
> **Correlational Scores↓**：
> | Model         | Sines          | Electricity     | ETTh1          | Energy          |
> |---------------|----------------|-----------------|----------------|-----------------:|
> | CPDD          | 0.329±.005   | 0.198±.001  | 0.247±.003   | **1.912±.001**  |
> | Diffusion-TS  | **0.194±.003** | 0.213±.001    | 0.430±.008     | 7.271±.007      |
> | Timegan       | 1.392±.003     | 0.726±.002      | 1.811±.001     | 15.581±.003     |
> | Timevae       | 4.263±.001     | **0.086±.001**  | **0.155±.004** | 2.484±.004    |
>
> Overall, CPDD demonstrates excellent generation performance, excelling in both Context-FID and Correlational metrics, while also maintaining a good balance between Context consistency and multivariate time series correlation.
>
> >**Q 2**： In Table 2, in the Predictive Score metric, the model with DSConv removed achieves better performance in the Sines dataset, and the model with TPC removed exhibits the best performance in the Energy dataset. The results of the ablation experiments are puzzling, which may shake the rationality of the structural design of DSConv and TPC.
>
> Thanks for the comment of the reviewer. We choose the combination structure with the best generalization for each dataset. Specifically, the role of DSConv is closely linked to the dataset's complexity, while the role of TPC is associated with the chosen compression ratio.
>
> The Sines dataset and the Energy dataset are two significantly different datasets. In the Sines dataset, the frequency and phase of the sine wave function for each channel are obtained through random sampling. Its periodicity and relatively simple structure can not fully leverage the capabilities of DSConv to effectively capture complex multivariate dependencies. The Energy dataset represents energy consumption inside buildings and indoor/outdoor temperature and light levels for each channel, exhibiting complex patterns and noise.
>
> Moreover, as the experiment utilized all 28 channel dimensions of Energy, we set the feature dimension for compression encoding to 32 to save memory. The compression ratio is as high as (1024th×28)/(64th×32)=14. For devices with sufficient memory, CPDD can utilize a lower compression ratio, typically around 2 to 4 times, to decrease the Predictive Score and achieve a superior generation effect.
>
>  Additionally, during the experiment, we observed that in long-time series generation, as the channel dimension increases, the Discriminative score metric tends to be higher. As shown in the table below, the TPC model trained on the traffic dataset was used to test the reconstructed time series output against the original time series, with different channels randomly selected.
>
> | Dataset  | Dim=2       | Dim=10      | Dim=30      | Dim=50      |
> |----------|-------------|-------------|-------------|-------------|
> | Traffic ↓ | 0.034±0.005 | 0.202±0.046 | 0.496±0.004 | 0.499±0.001 |
>
> The TPC model trained on the Traffic dataset has a reconstruction MSE loss of 0.0869, with the original time series length of 1024th, channel dimension of 50, compression encoding length of 64th, and feature dimension of 32. In the case of training data with long-time series and high channel dimensions, the classification model may experience overfitting.

---

> ### Author Response · Authors · 2024-11-22
> **Reference**
>
> [1] XinyuYuanandYanQiao.Diffusion-TS:Interpretablediffusionforgeneraltimeseriesgeneration.
>  InTheTwelfthInternationalConferenceonLearningRepresentations,2024.
>
> [2] Jeha Paul, Bohlke-Schneider Michael, Mercado Pedro, Kapoor Shubham, Singh Nirwan Rajbir, Flunkert Valentin, Gasthaus Jan, and Januschowski Tim. Psa-gan: Progressive self attention gans for synthetic time series, 2022.
>
> [3] Hao Ni, Lukasz Szpruch, Magnus Wiese, Shujian Liao, and Baoren Xiao. Conditional sig wasserstein gans for time series generation. arXiv preprint arXiv:2006.05421, 2020.

---

> > ### Comment · Reviewer_fwsP · 2024-11-27
> > **Thank you for your response**
> >
> > Thank you for your response. I appreciate the efforts, and it seems some of my concerns were not well addressed. In consideration of baselines, you may not only compare with diffusion models, but also consider other  strong deep learning baselines that also designed for time series forecasting. Furthermore, it seems the proposed method lacks of scalability in high dimensional series. I will remain my scores.

---

> ### Author Response · Authors · 2024-11-27
> **Response to Reviewer fwsP**
>
> >In consideration of baselines, you may not only compare with diffusion models, but also consider other strong deep learning baselines that also designed for time series forecasting.
>
> Thank you to the reviewers for their time and feedback. We believe there may be a misunderstanding between **generative models** and **general-purpose generative models**（such as Timer，TimeMixer）. Our work focuses on generative models specifically designed for **multivariate long time series generation**, which differ fundamentally from autoregressive models or general-purpose large models commonly used in **forecasting**. Therefore, a comparison with forecasting models is not applicable to our approach, and we maintain that the specialized domain of time series generation remains significant for further research.
>
> >Furthermore, it seems the proposed method lacks of scalability in high dimensional series.
>
> Thank you for bringing up these concerns about scalability.  We want to point out that the simultaneous handling of high channel dimensions (300-800) and long sequences (1024th) in time series generation is actually an unexplored frontier in current research.  After thorough literature review, we haven't found any previous work that successfully tackles generation tasks of this scale and complexity.  This highlights both the challenging nature of our undertaking and its potential significance for advancing the field.  We would welcome any references to works that have achieved similar scalability in time series generation, as this would greatly benefit our research direction.

---

### Official Review · Reviewer_afTP · 2024-10-29

**Soundness:** 2
**Presentation:** 2
**Contribution:** 2
**Rating:** 3
**Confidence:** 4

**Summary:**

This paper introduces CPDD, a method for time series generation, which addresses challenges in balance between long-term dependencies and short-term feature learning. It utilizes a patch Compressed module based on the patch mode decomposition method to obtain the latent encoding of multi-scale feature of time series. It utilizes a diffusion-based model to learn the latent distribution and decode the resulting samples, which achieves state-of-the-art performance in the generation task of multivariate long-time series with efficiency and robustness.

**Strengths:**

1. Methodology: CPDD is a patch compression method to capture complex long-term and short-term dependencies more effectively with the diffusion model for high-quality samples generation.
2. Empirical Results: The numerical results presented in the paper are compelling, showing significant improvements over competing
approaches in terms of generated time series quality. This empirical evidence supports the effectiveness of the proposed method.
3. Proofs: The author gives a detailed formal proof of the effectiveness of DSConv blocks' structural regularization.

**Weaknesses:**

1.  The problem that the article aims to address is confusing; what does "balance between long-term dependencies and short-term feature learning" mean?
2. In line 25 of the article, the author mentions "efficiency." How is this demonstrated in the article? Please compare the model's memory usage and inference time.
3. In the embedding space shown in Figure 1, what do "distant" and "nearby" mean? This is quite confusing.
4. The entire CPDD process is quite confusing. please provide a specific implementation process or corresponding pseudocode?
5. In lines 340-341 of the article, the author mentions "Z: latent representation obtained from the TPC Encoder during training." Then why is there no loss term for the TPC Encoder/Decoder in Equation 16? Is CPDD an end-to-end or a two-stage process? Please provide a detailed explanation.
6. What is “L_{AFC}" in Equation 16? Is there any difference between "L_{AFC}" and "L_{ACF}"?
7. The writing issues in the article are evident, with many sentences being difficult to understand, and the challenge that article aims to address is not clearly defined.

**Questions:**

Please refer to the Weaknesses.

---

> ### Author Response · Authors · 2024-11-22
> **Response to Reviewer afTP**
>
> We deeply appreciate the reviewer's thorough evaluation and thoughtful feedback. The suggestions have been instrumental in improving the clarity, rigor, and overall quality of our work.
>
> We have thoroughly reviewed the valuable feedback from the reviewers and have revised the manuscript accordingly. Detailed responses to each comment are provided below for clarity and transparency.
>
> >**W 1**: The problem that the article aims to address is confusing; what does "balance between long-term dependencies and short-term feature learning" mean?
>
> We apologize for not clearly articulating the concept of "balancing the learning of long-term dependencies and short-term features".  Generally, the long-term dependencies in time series data manifest as trend changes and cyclical fluctuations, while short-term features reflect local variations or instantaneous mutations. Common methods in deep learning to learn long-term dependencies and short-term features involve multiscale feature fusion and multilevel structures. The paper employs a similar multilevel structure approach, initially compressing through TPC into a combination of long-term, short-term, and residual modal functions for representation encoding, and then utilizing a diffusion model to learn the distribution of representation encoding in latent space.
>
> >**W 2**: In line 25 of the article, the author mentions "efficiency." How is this demonstrated in the article? Please compare the model's memory usage and inference time.
>
> We apologize for not clearly articulating this advantage and providing the corresponding analysis on "efficiency." This aspect primarily pertains to models using attention or Transformer. Our approach incorporates a multiscale feature extraction embedding layer, significantly reducing the number of input tokens while slightly increasing the feature dimension. The input token sequence length L is shortened from 1/16 to 1/4, with the feature dimension d doubling to quadrupling. The computational complexity of the Transformer is $\mathcal{O}(d L^2 + Ld^2)$. When the sequence length dimension is significantly larger than the feature dimension, the computational complexity effectively decreases. The specific training and inference time comparative results are presented in the table below:
>
> | Dataset       | Model         | Training time (min) | Sample time (min) |
> |:-------------:|:-------------:|:-------------------:|:-----------------:|
> | Sines         | Diffusion-TS  | 146                 | 65                |
> |               | CPDD          | **63**                  | **30**                |
> | Electricity   | Diffusion-TS  | 109                 | 495               |
> |               | CPDD          | **92**                  | **65**                |
> | Etth1         | Diffusion-TS  | 181                 | 111               |
> |               | CPDD          | **74**                  | **42**                |
> | Energy        | Diffusion-TS  | 185                 | 410               |
> |               | CPDD          | **128**                 | **38**                |
>
> The shorter training and sampling times of CPDD compared to Diffusion-TS indicate that CPDD effectively reduces the computational complexity.
>
> >**W 3**: In the embedding space shown in Figure 1, what do "distant" and "nearby" mean? This is quite confusing.
>
> We apologize for not making this clear. The terms "nearby" and "distant" refer to the magnitude of the Euclidean distance between two embedding vectors. Specifically, "nearby" indicates a small distance between two embedding vectors, implying that the time series segments corresponding to these vectors exhibit higher similarity in terms of features or patterns learned by the model; whereas "distant" signifies a large distance between two vectors, indicating significant differences in features or patterns for the corresponding time series segments.

---

> ### Author Response · Authors · 2024-11-22
> **Response to Reviewer afTP**
>
> >**W 4**: The entire CPDD process is quite confusing. please provide a specific implementation process or corresponding pseudocode?
>
> We sincerely apologize for not prominently providing an overall training process. The pseudo-code for the training process of the two stages of CPDD is presented as:
>
> | **Algorithm: Two-Stage Training Framework for Time-series Patch Compressed (TPC) Module and Diffusion Module** |
> |--------------------------------------------------------------------|
>
> ### **Stage I: TPC Pretraining**
> 1. **Encode input**:
>    $ Z \leftarrow TPCEncoder(X)   $
> 2. **Decode reconstruction**:
>   $  \hat{X} \leftarrow TPCDecoder(Z)   $
> 3. **Compute reconstruction loss**:
>   $
>    \text{Loss} = || X - \hat{X} ||^2
>    $
> 4. **Optimize TPCEncoder and TPCDecoder parameters using gradient descent.**
>
> ---------------------------------------------------
>
> ### **Stage II: Diffusion Training**
> 1. **Initialization**: Freeze TPCEncoder after pretraining:
>   $  Z_0 \leftarrow TPCEncoder_{\text{freeze}}(X)   $
> 3. **Generate noisy latent representation**:
>
>    $Z_t = \sqrt{\bar{\alpha}_t} Z_0 + \sqrt{1 - \bar{\alpha}_t} \varepsilon,$
>
> where  $\bar{\alpha}_t = \alpha_1 \alpha_2 \dots \alpha_t, \quad \alpha_t = 1 - \beta_t $.
>
> 4. **Predict original latent representation**:
>    $ \hat{Z}_0 \leftarrow Diffusion(Z_t, t) $.
> 5. **Compute total loss**:
>
>    $\text{Loss} = \lambda_1 \alpha_t  \frac{(1 - \bar{\alpha}_t)}{\beta_t^2} || Z_0 - \hat{Z}_0 ||^2 $
>  $+ \lambda_2 \sum_{m=1}^M w(m) | \hat{r}(m) - r(m) | + \lambda_3 || X - \hat{X} ||^2.$
>
> 6. **Optimize Diffusion model parameters using gradient descent.**
>
> >**W 5**: In lines 340-341 of the article, the author mentions "Z: latent representation obtained from the TPC Encoder during training." Then why is there no loss term for the TPC Encoder/Decoder in Equation 16? Is CPDD an end-to-end or a two-stage process? Please provide a detailed explanation.
>
> We apologize for not explicitly listing the reconstruction loss function used for TPC training due to an organizational error in writing. We have added the loss function of TPC ($\mathcal{L}_{recon}=||X-\hat{X}||^2$ ) in the paper. As the weakness 4 answers, CPDD is a two-stage process. Initially, we trained the TPC as the latent encoder, followed by utilizing the TPC encoding with fixed weights to acquire the latent encoding for training the diffusion model. Subsequently, the latent code produced by the diffusion model is decoded by TPC decoder to generate the output, completing the generation of the final multivariate long-time series.
>
> >**W 6**: What is “L_{AFC}" in Equation 16? Is there any difference between "L_{AFC}" and "L_{ACF}"?
>
> Thanks to the reviewer for the careful review, AFC is a writing error and only ACF is the correct notation. $\mathcal{L}_{ACF}$ represents the autocorrelation loss, which is used to guide the model to learn the autocorrelation structure of the input time series.

---

> > ### Author Response · Authors · 2024-11-22
> > **Response to Reviewer afTP**
> >
> > >**W 7**: The writing issues in the article are evident, with many sentences being difficult to understand, and the challenge that article aims to address is not clearly defined.
> >
> > We apologize for not using more easily understandable sentences due to our lack of writing skills. To better articulate the ideas and methods of the paper, we will seek guidance from more experienced researchers to enhance the organization and expression of the paper. This study aims to address the key challenges in generating multi-dimensional long-time series, focusing on the problem of generating multi-dimensional time series with lengths exceeding 256th. This issue remains unexplored in existing research, especially in complex domains such as electricity and transportation, where there is a high demand for long-time series generation, yet the performance of existing methods is still limited.
> >
> > Multi-dimensional long-time series generation not only requires generating long-time series with high fidelity but also maintaining complex dynamic patterns in the generated data and dependencies between variables. Taking the electricity and transportation sectors as examples, analytical tasks often rely on long-span time series data to capture seasonal trends, periodic fluctuations, and interactive patterns among multiple variables. However, existing generation models (such as frameworks based on autoregressive models, generative adversarial networks, or diffusion models) mostly focus on data sequences of lengths not exceeding 256th, making it challenging to effectively extend to longer sequences, especially in multi-variable settings.
> >
> > We believe that the challenge in generating multi-dimensional long-time series lies in existing methods' inability to simultaneously consider long-term dependencies and short-term feature learning. Therefore, we explored a method to effectively compress and encode multi-dimensional long-time series into shorter time series. This method utilizes a time series patch mode decomposition technique to break down patches into long-term, short-term, and residual mode functions, effectively preserving long-term dependencies and short-term feature information in the compression encoding, thereby laying the foundation for the potential diffusion generation in the second stage. We will promptly make the revision in the manuscript.

---

> > > ### Comment · Reviewer_afTP · 2024-11-27
> > > **Thank you for your response**
> > >
> > > Thank you for your response, which has addressed most of my concerns. However, I still cannot agree with the challenge that CPDD aims to address. In both the submitted paper and the responses provided by the authors, the meaning of “balance between long-term dependencies and short-term feature learning” has not been adequately addressed. The authors continue to mention that CPDD employs a decoupled approach to learning long- and short-term multi-scale features in time series, but this idea is quite common in time series domain, as demonstrated by works such as Scaleformer [1], Moirai [2], etc. Therefore, I keep my score.
> > >
> > > [1]. Shabani, Mohammad Amin, et al. "Scaleformer: Iterative Multi-scale Refining Transformers for Time Series Forecasting." The Eleventh International Conference on Learning Representations. (ICLR2023)
> > >
> > > [2]. Woo, Gerald, et al. "Unified Training of Universal Time Series Forecasting Transformers." Forty-first International Conference on Machine Learning. (ICML2024)

---

> ### Author Response · Authors · 2024-11-27
> **Thank you for your response**
>
> >However, I still cannot agree with the challenge that CPDD aims to address. In both the submitted paper and the responses provided by the authors, the meaning of “balance between long-term dependencies and short-term feature learning” has not been adequately addressed.
>
> Thank you for raising this important question about our method's advantages.
> CPDD achieves an effective balance between long-term dependencies and short-term features through a **divide-and-conquer approach**.  By dividing the time series into patches, we decompose the complex pattern learning problem into two distinct sub-tasks: local feature (short-term feature) learning within patches and global dependency (long-term dependency) learning between patches.  These two levels of features are processed independently using specialized components (Transformer for inter-patch relationships and DSConv for intra-patch patterns) and then merged into a unified latent space representation.  This hierarchical processing strategy enables more efficient and focused learning at different temporal scales, representing an innovative approach not found in existing temporal generation methods.
>
> >The authors continue to mention that CPDD employs a decoupled approach to learning long- and short-term multi-scale features in time series, but this idea is quite common in time series domain, as demonstrated by works such as Scaleformer [1], Moirai [2], etc. Therefore, I keep my score.
>
> We appreciate your careful examination of our method's novelty. We'd like to clarify a fundamental difference between CPDD and existing multi-scale approaches in time series analysis:
> The key distinction lies not in whether we use multi-scale processing, but in why and how we apply it. While Scaleformer and Moirai utilize multi-scale architectures primarily for feature extraction and representation, CPDD approaches the problem from an **optimization perspective**. Our method specifically addresses the computational and optimization challenges in temporal pattern generation through **structured decomposition**:
>
> Problem Formulation:
>
>
> Traditional multi-scale methods focus on learning better representations at different time scales
> CPDD, however, uses decomposition to solve the inherent optimization conflict between capturing local patterns and modeling long-range dependencies.
>
>
> Solution Strategy:
>
>
> Instead of parallel multi-scale feature extraction, CPDD employs a structured divide-and-conquer approach to break down the learning objective.
> This decomposition significantly reduces the optimization difficulty by allowing specialized components to focus on distinct aspects of the temporal dynamics.
>
>
> Generation Focus:
>
>
> While existing methods excel at feature extraction for tasks like classification and prediction, CPDD is specifically designed for the unique challenges of temporal pattern generation.
> Our approach enables more controlled and efficient pattern synthesis by separately handling local details and global structure.
>
> This fundamental difference in problem formulation and solution strategy distinguishes CPDD from existing multi-scale approaches. We apologize for not making this distinction clearer in the paper and would appreciate the opportunity to better articulate these differences.

---

### Official Review · Reviewer_F7YS · 2024-10-29

**Soundness:** 2
**Presentation:** 2
**Contribution:** 2
**Rating:** 5
**Confidence:** 4

**Summary:**

This paper presents a diffusion-based method for time series generation that integrates a patch compression module with trend-seasonal decomposition to enhance generation performance.

**Strengths:**

1. This paper effectively leverages a patch compression method to capture complex long-term and short-term dependencies in time series data.

2. The authors employ trend-seasonal decomposition to facilitate the diffusion model's ability to learn complex distribution shifts.

**Weaknesses:**

1. The evaluation experiments presented in the paper are insufficient to convincingly demonstrate the effectiveness of the proposed method. Specifically, more common used evaluation metrics need to be added (like MSE, MAE, .etc), and the selection of baseline methods (both the diffusion-based methods and the transformer-based methods should be compared) and datasets is not comprehensive enough to provide a robust comparison.

2. The formulation of the paper needs significant improvement; the organization and clarity of the text make it difficult to identify the key ideas and contributions.

3. The integration of the proposed patch compression method with seasonal-trend decomposition seems to offer limited novelty, as this combination may be viewed as a relatively minor contribution to the existing body of work in this area.

4. While the paper employs a transformer as the encoder within the diffusion model, it is essential to consider the associated computational costs when making comparisons with baseline methods.

**Questions:**

1. How are the short-term and long-term modes decomposed from the patches?

2. What is the rationale for employing the transformer as the encoder of the diffusion model instead of using the transformer directly?

---

> ### Author Response · Authors · 2024-11-22
> **Response to Reviewer F7YS**
>
> We are grateful to the reviewer for their detailed and insightful comments, which have provided us with an opportunity to enhance the presentation and technical soundness of our paper. We greatly value the reviewers’ thoughtful comments, which have guided us in improving our work. To address these concerns, we provide detailed responses to each point below.
>
> >**W 1**: The evaluation experiments presented in the paper are insufficient to convincingly demonstrate the effectiveness of the proposed method. Specifically, more common used evaluation metrics need to be added (like MSE, MAE, .etc), and the selection of baseline methods (both the diffusion-based methods and the transformer-based methods should be compared).
>
> We thank the reviewer's insightful suggestion. In the first point, we address the evaluation experiments question, and in the second point, we discuss the baseline methods.
> 1. In generation tasks, there are usually multiple valid outputs, but MAE and MSE calculations are based on point-to-point errors and do not account for the diversity or semantic similarity between the generated results and the reference targets. We have added the Context-FID[1] and Correlational[2] evaluation metrics in the table below, which are more suitable for evaluating generative tasks as they account for the semantic consistency and distributional similarity of the generated outputs in the table below. Context-FID evaluates the overall quality of the generated time series by extracting features using a pre-trained embedding model and measuring the distributional differences between generated and real sequences in the embedding space. Correlational assesses whether the generated data preserves the statistical structure and dependency patterns of the time series by comparing the auto-correlation and cross-correlation distributions with those of the real data.
>
> **Context-FID Scores↓**:
> | Model         | Sines          | Electricity     | ETTh1          | Energy          |
> |---------------|----------------|-----------------|----------------|-----------------:|
> | CPDD          | **5.687±.252** | 5.996±.294    | **3.238±.421** | **1.151±1.072** |
> | Diffusion-TS  | 13.451±.492  | 65.204±1.853    | 20.568±2.973   | 67.630±7.732    |
> | Timegan       | 59.031±2.223   | 18.365±.613     | 10.381±1.227   | 61.022±1.893  |
> | Timevae       | 106.981±.722   | **4.804±.436**  | 3.362±.432   | 19.862±.038     |
>
> **Correlational Scores↓**：
> | Model         | Sines          | Electricity     | ETTh1          | Energy          |
> |---------------|----------------|-----------------|----------------|-----------------:|
> | CPDD          | 0.329±.005   | 0.198±.001  | 0.247±.003   | **1.912±.001**  |
> | Diffusion-TS  | **0.194±.003** | 0.213±.001    | 0.430±.008     | 7.271±.007      |
> | Timegan       | 1.392±.003     | 0.726±.002      | 1.811±.001     | 15.581±.003     |
> | Timevae       | 4.263±.001     | **0.086±.001**  | **0.155±.004** | 2.484±.004    |
>
> Overall, CPDD demonstrates excellent generation performance, excelling in both Context-FID and Correlational metrics, while also maintaining a good balance between Context consistency and multivariate time series correlation.
>
> 2. In the experiment of the paper, we have compared Diffusion-TS, which is based on the Diffusion model and Transformer block. We acknowledge the importance of comparing more Diffusion-based and Transformer-based methods. Nevertheless, we are still in the process of conducting additional baseline tests due to time limitations. We will expedite the completion of all baseline tests and promptly provide updates in the comments section or appendix of the paper.

---

> ### Author Response · Authors · 2024-11-22
> **Response to Reviewer F7YS**
>
> >**W 2**: The formulation of the paper needs significant improvement; the organization and clarity of the text make it difficult to identify the key ideas and contributions.
>
> We sincerely apologize for our poor expression. We will promptly revise the relevant statements in the paper. Allow us to attempt to introduce the design context of CPDD in the following. In order to address the challenge of generating multivariate long-time series, we seek to extend the method of image latent diffusion generation to the temporal domain. However, despite the excellent performance of the VAE latent encoder in image latent generation, it faces the following issues in temporal latent diffusion generation:
> 1. Insufficient temporal dependency: VAE and other temporal latent encoders cannot effectively capture both long-term dependencies and short-term features.
> 2. Limited latent space distribution: In temporal diffusion, the uneven distribution of data (such as low sample density in certain time periods) may lead to uneven representations in the latent space initialized by VAE and other latent encoders, potentially causing the diffusion process to favor generating samples in high-density areas.
> 3. High latent encoding dimension: Unlike the high redundancy of image information, temporal information has lower redundancy. To achieve high-quality generation, VAE and other latent encoders may require a very high latent dimension in the initialized latent space.
>
> Therefore, our core idea is to seek a novel latent encoder that can generalize the representation of multivariate long-time series with a lower latent dimension while effectively preserving their temporal dependency. This encoder should possess the ability to efficiently compress a wide range of temporal data and maintain dynamic features in the encoding rather than solely static representations. Combining the above analysis, our contribution lies in proposing the temporal Patch modal decomposition technique to meet the aforementioned requirements and introducing the framework of CPDD, achieving high-quality generation of multivariate long-time series.
>
> >**W 3**: The integration of the proposed patch compression method with seasonal-trend decomposition seems to offer limited novelty, as this combination may be viewed as a relatively minor contribution to the existing body of work in this area.
>
> Thanks the reviewer’s constructive feedback regarding the integration of the patch compression method with seasonal-trend decomposition. We appreciate the perspective and will further emphasize how this combination contributes to addressing specific challenges in long-term time series generation, as well as clarify its novelty compared to existing methods.
> We acknowledge that the approach of time-series mode function decomposition is not entirely novel and that traditional time series analysis methods such as VMD[3] and EMD[4] effectively utilize this concept.
>
> However, the integration of mode function decomposition with deep learning for enhanced time series analysis remains a relatively unexplored direction. Specifically, EMD and VMD methods lack the capability to parallelly decompose time series using GPU, hindering direct integration with deep learning techniques. In contrast, the DSConv module introduced in this study facilitates the model in learning a series of Patch mode functions to achieve a more comprehensive representation.
>
> CPDD utilizes time series patches as the fundamental unit for decomposition, aiming to simplify the process of VMD or EMD. Each time series patch is a composition of long-term, short-term, and residual patterns. Limiting the patch length to a range of 4 to 16 enables us to efficiently employ processing components for long-term, short-term, and residual modes, leading to successful decomposition outcomes. Employing Time-series Patch mode function decomposition technology results in a compression ratio ranging from 1 to 14 times (the original data volume/compressed data volume). This approach outperforms the conventional deep learning-based time series compression encoder in achieving superior compression outcomes[5-6].

---

> ### Author Response · Authors · 2024-11-22
> **Response to Reviewer F7YS**
>
> >**W 4**: While the paper employs a transformer as the encoder within the diffusion model, it is essential to consider the associated computational costs when making comparisons with baseline methods.
>
> Although our model incorporates Transformer, the dimensions of its input tokens are altered after passing through the embedding layer. The input token sequence length L is reduced to 1/16 to 1/4, while the feature dimension $D$ is doubled to quadrupled. The computational complexity of the Transformer is $\mathcal{O}(DL^2 + LD^2)$. When $L$ is significantly larger than $D$, the computational complexity of CPDD effectively decreases. We conducted comparative tests to supplement the model's training and inference efficiency, with the results presented in the table below.
>
> | Dataset       | Model         | Training time (min) | Sample time (min) |
> |:-------------:|:-------------:|:-------------------:|:-----------------:|
> | Sines         | Diffusion-TS  | 146                 | 65                |
> |               | CPDD          | **63**                  | **30**                |
> | Electricity   | Diffusion-TS  | 109                 | 495               |
> |               | CPDD          | **92**                  | **65**                |
> | Etth1         | Diffusion-TS  | 181                 | 111               |
> |               | CPDD          | **74**                  | **42**                |
> | Energy        | Diffusion-TS  | 185                 | 410               |
> |               | CPDD          | **128**                 | **38**                |
>
> The shorter training and sampling times of CPDD compared to Diffusion-TS indicate that CPDD effectively reduces the computational complexity.
>
> >**Q 1**: How are the short-term and long-term modes decomposed from the patches?
>
> We are grateful to the reviewers for enabling us to improve this point.
>
> The spontaneous decomposition of time series into long-term and short-term patterns is an inherent characteristic of the model's structure. While the Time-series Patch compressed (TPC) module leverages the moving average decomposition technique, its primary impact lies in accelerating training convergence.
>
> The design of DSConv emphasizes temporal feature learning within the patch, facilitating the identification of short-term patterns. While the ConvFFN2 component targets the boundary information of adjacent patches and single-channel convolution emphasizes the relative scale information of patches across a broader range, the narrow channel design creates an information bottleneck that hinders the excessive development of non-short-term patterns.
>
> DSConv introduces the StarNet[7] that explicitly creates a latent space. In this structure, the ConvFFN output acts as the base of this space, and the single-channel convolution output serves as the coefficient within this latent space. This setup constrains the patterns learned by DSConv to focus on the localized, short-term temporal features within the patches. The StarNet structure enforces a form of hierarchical decomposition, emphasizing localized dynamics while suppressing excessive cross-patch interference.
>
> In contrast, Transformer excels in capturing long-distance dependencies and global patterns due to its superior global attention mechanism, giving it a notable edge in modeling long-term features.
>
> Hence, the integration of DSConv and Transformer can establish structural regularization, facilitating the autonomous separation of modes across the long-term and the short-term.
>
> >**Q 2**: What is the rationale for employing the transformer as the encoder of the diffusion model instead of using the transformer directly?
>
> Thanks to the reviewer for this insightful question regarding the rationale behind employing the transformer as the encoder in the diffusion model. In the context of time series generation tasks, training data is often scarce, and using Transformer directly as a generation model can lead to overfitting. Utilizing a Transformer encoder can enhance global feature extraction capabilities, facilitating the capture of long-term patterns in the time series.

---

> ### Author Response · Authors · 2024-11-22
> **Reference**
>
> >**Reference**:
>
> [1] Jeha Paul, Bohlke-Schneider Michael, Mercado Pedro, Kapoor Shubham, Singh Nirwan Rajbir, Flunkert Valentin, Gasthaus Jan, and Januschowski Tim. Psa-gan: Progressive self attention gans for synthetic time series, 2022.
>
> [2] Hao Ni, Lukasz Szpruch, Magnus Wiese, Shujian Liao, and Baoren Xiao. Conditional sig wasserstein gans for time series generation. arXiv preprint arXiv:2006.05421, 2020.
>
> [3]K. Dragomiretskiy and D. Zosso, "Variational Mode Decomposition," in IEEE Transactions on Signal Processing, vol. 62, no. 3, pp. 531-544, Feb.1, 2014, doi: 10.1109/TSP.2013.2288675.
>
> [4]Huang N E, Shen Z, Long S R, et al. "The empirical mode decomposition and the Hilbert spectrum for nonlinear and non-stationary time series analysis." Proceedings of the Royal Society of London. Series A: mathematical, physical and engineering sciences 454.1971 (1998): 903-995.
>
> [5]Ranak Roy Chowdhury, Xiyuan Zhang, Jingbo Shang, Rajesh K Gupta, and Dezhi Hong. Tarnet: Task-aware reconstruction for time-series transformer. In Proceedings of the 28th ACM SIGKDD Conference on Knowledge Discovery and Data Mining, pp. 212–220, 2022.
>
> [6]George Zerveas, Srideepika Jayaraman, Dhaval Patel, Anuradha Bhamidipaty, and Carsten Eick hoff. A transformer-based framework for multivariate time series representation learning. In Proceedings of the 27th ACM SIGKDD Conference on Knowledge Discovery & Data Mining, KDD ’21, pp. 2114–2124, New York, NY, USA, 2021. Association for Computing Machinery. ISBN 9781450383325.
>
> [7] Xu Ma, Xiyang Dai, Yue Bai, Yizhou Wang, and Yun Fu. Rewrite the stars. In Proceedings of the
> IEEE/CVF Conference on Computer Vision and Pattern Recognition, 2024.

---

### Official Review · Reviewer_tQBV · 2024-11-04

**Soundness:** 3
**Presentation:** 3
**Contribution:** 3
**Rating:** 6
**Confidence:** 4

**Summary:**

This paper aims to address the challenge of balancing the long-term dependencies and short-term features in time series generation and proposes a novel model named Compressed Patch Denoising Diffusion-model (CPDD). The proposed approach first employs a Time-series Patch Compression (TPC) module to decompose time series patches into mode functions, effectively capturing latent representations of both long-term and short-term features. Afterward, a diffusion-based model with a CNN backbone is designed to learn the latent distributions and generate multivariate long-term time series. Experimental results demonstrate that CPDD achieves the SOTA performance in time series generation. Furthermore,  the robustness and generalization capabilities of the TPC module are rigorously verified.

**Strengths:**

- The generative modeling approach of decomposing time series patches into mode functions presented in this paper is novel and well-positioned in the literature on time series generation, to the best of my knowledge.
- The introduction of the Time-series Patch Compression (TPC) module marks a notable innovation in time series modeling. This module provides a robust alternative to the commonly used autoencoder-based compression methods and trend-seasonality decomposition techniques. The exploration of its robustness and generalization is particularly noteworthy
- Figures 1 and 2 provide a clear illustration of the proposed approach, making the core designs easier to understand.

**Weaknesses:**

- This paper identifies high computational demands as a limitation of existing methods, but the proposed approach also employs a computationally intensive Transformer-based architecture. Therefore, a detailed analysis of the computational complexity of the proposed CPDD is essential.
- The visualizations in this paper are generally of high quality. Unifying the font size in Figure 4, especially on the left-side module, to match that of other figures would improve visual consistency and readability.

**Questions:**

1. CPDD divides the entire time series into N patches, which might pose a risk of disrupting critical temporal patterns at the boundaries of these patches. Could this potentially affect the model's ability to accurately capture and reproduce these dynamics?
2. How does the patch length N impact the model performance? Is there an optimal range of N?
3. Can you provide a detailed comparison of the computational complexity between CPDD and baselines? For example, a table comparing their training and inference time.

---

> ### Author Response · Authors · 2024-11-22
> **Response to Reviewer tQBV**
>
> We sincerely thank the reviewer for their time, effort, and valuable feedback on our work. We are very grateful to the reviewer for their recognition of our work, which encouraged us to further improve the quality and clarity of the manuscript. The insightful comments and constructive suggestions have significantly helped us identify areas for improvement and further refine our manuscript.
>
> We have carefully considered the reviewer's comments and made corresponding revisions to the manuscript. Below, we address each point raised by the reviewer in detail.
>
> >**W1**: Therefore, a detailed analysis of the computational complexity of the proposed CPDD is essential.
>
> >**Q3**: Can you provide a detailed comparison of the computational complexity between CPDD and baselines? For example, a table comparing their training and inference time.
>
> Thanks for the reviewer's insightful suggestion. Despite CPDD utilizing Transformer block, the dimension of input tokens undergoes changes post the embedding layer. Here, the token sequence input length $L$ will decrease to 1/16 to 1/4 of its original size, while the feature dimension $D$ will increase to 2 to 4 times its initial scale. The computational complexity of Transformer is $\mathcal{O}(DL^2+LD^{2})$. When the sequence length dimension is much larger than the feature dimension, the computational complexity will be effectively reduced. The specific comparison of training time and inference time is shown in the following table:
>
> | Dataset       | Model         | Training time (min) | Sample time (min) |
> |:-------------:|:-------------:|:-------------------:|:-----------------:|
> | Sines         | Diffusion-TS  | 146                 | 65                |
> |               | CPDD          | **63**                  | **30**                |
> | Electricity   | Diffusion-TS  | 109                 | 495               |
> |               | CPDD          | **92**                  | **65**                |
> | Etth1         | Diffusion-TS  | 181                 | 111               |
> |               | CPDD          | **74**                  | **42**                |
> | Energy        | Diffusion-TS  | 185                 | 410               |
> |               | CPDD          | **128**                 | **38**                |
>
> The shorter training and sampling times of CPDD compared to Diffusion-TS indicate that CPDD effectively reduces the computational complexity.
>
> >**W 2**: The visualizations in this paper are generally of high quality. Unifying the font size in Figure 4, especially on the left-side module, to match that of other figures would improve visual consistency and readability.
>
> Thank the reviewer for highlighting this issue. We agree that unifying the font size in Figure 4, especially in the left-side module, will improve visual consistency and readability. We have modified figure 4 and ensured all figures are consistent in style and presentation in the paper.
>
> >**Q 1**: CPDD divides the entire time series into N patches, which might pose a risk of disrupting critical temporal patterns at the boundaries of these patches. Could this potentially affect the model's ability to accurately capture and reproduce these dynamics?
>
> We thank the comment of the reviewer. To address the potential boundary effect issue due to patch segmentation, we implement **two** crucial measures for effective mitigation:
>
> 1. Inheriting the overlapping patch strategy from PatchTST[1]. Concretely, by configuring a suitable stride to create overlapping regions between neighboring patches, this approach significantly minimizes information fragmentation at the segmentation boundary while preserving the coherence of local temporal features.
>
> 2.  Additionally, within the model architecture, the ConvFFN2 submodule is uniquely crafted within our DSConv module. This submodule focuses on learning the temporal correlations among adjacent patches and reviving temporal dynamic features that could be compromised by patch segmentation.
>
> Overall, boundary effects and the preservation of temporal dynamics have been mitigated through overlapping patch segmentation and the ConvFFN2 submodule.
>
> >**Q 2**: How does the patch length N impact the model performance? Is there an optimal range of N?
>
> We thank the reviewer for the comments, which we lack a detailed description in the paper. The selection of the Patch length N correlates with the input time series length L, the channel dimension C, and the model's hidden dimension. Consequently, a longer Patch length and a larger channel dimension imply a higher potential for diverse underlying mode functions, necessitating a larger hidden dimension. In the experiment, the hyperparameters for the Time-series patch Compressed (TPC) module are defined as follows: for input lengths ranging from the 64th to the 1024th and channel dimensions between 1 and 50, the Patch length N is set between 4 and 16, with token hidden dimensions varying from 128 to 256.

---

> > ### Comment · Reviewer_tQBV · 2024-11-24
> >
> > Thanks for your responses. I appreciate the clarifications and revisions provided. Most of my concerns have been addressed. However, there are still a few points that require further explanation:
> >
> > - What are the lengths of time series used in the updated comparisons of training time and inference time? Please specify the data settings for the empirical computational costs.
> >
> > - My concerns regarding the effect of patch length N on model performance have not been fully addressed, particularly the first part of my question 2. Specifically, is the model's performance consistent across various patch lengths within the possible range? Would it be possible that only certain finely tuned patch lengths yield the superior performance demonstrated in the experiments, while other patch lengths do not maintain this level of effectiveness? Diving into the settings in your response, does the model perform stably across the patch lengths ranging from 4 to 16?
> >
> >
> > Furthermore, the clarified settings (e.g., overlapping patches) and updated results (e.g.. empirical computational costs) in the responses should be involved in the final version of this paper, preferably within an Appendix.

---

> > > ### Author Response · Authors · 2024-11-26
> > > **Response to Reviewer tQBV**
> > >
> > > I'm pleased to address the concerns you have raised.
> > > >What are the lengths of time series used in the updated comparisons of training time and inference time? Please specify the data settings for the empirical computational costs.
> > >
> > > The length of time series used in the updated training and inference time comparisons is 1024th with 7channels. We have now updated the detailed hyperparameters in Table 5 in section A.1 of the appendix.
> > >
> > > > My concerns regarding the effect of patch length N on model performance have not been fully addressed, particularly the first part of my question 2. Specifically, is the model's performance consistent across various patch lengths within the possible range? Would it be possible that only certain finely tuned patch lengths yield the superior performance demonstrated in the experiments, while other patch lengths do not maintain this level of effectiveness?
> > >
> > > Different patch size settings can yield good performance, but the range of patch sizes that exhibit the most balanced performance across various evaluation tests is relatively narrow. Given that the time series generation task necessitates evaluating the quality of generated data from multiple viewpoints, opting for a patch size that balances performance is justifiable. The comparison experiments of different patch sizes and their detailed hyperparameter settings have been revised and are now presented in Tables 6 and 7 in Part A.2 of the Appendix.
> > >
> > > >Diving into the settings in your response, does the model perform stably across the patch lengths ranging from 4 to 16?
> > >
> > > Based on the experimental results, the performance remains relatively stable. We can roughly estimate the optimal patch size setting and the corresponding feature dimension setting by considering the length and the number of channels in the input time series.
> > >
> > > We adhere to a specific hyperparameter setting rule to ensure that the compression ratio of TPC (length of input time series × number of channels / length of compressed time series × feature dimension) falls within the range of 1 to 4.  Some exceptions may arise, particularly when dealing with a high number of channels in the input time series, as demonstrated in Experiment 1 with 28 energy channels and a compression ratio of 14, resulting in inadequate memory allocation.

---

> > > > ### Comment · Reviewer_tQBV · 2024-11-27
> > > >
> > > > I appreciate your updates. My confusion has been clarified. Generally, this paper provides some new insights into multi-scale time series generation. I would like to rate it as "weakly accept", but no such option is provided. So, I will maintain my initial score.

---

> ### Author Response · Authors · 2024-11-22
> **Reference**
>
> >Reference:
>
> [1] Nie, Yuqi et al. “A Time Series is Worth 64 Words: Long-term Forecasting with Transformers.” ICLR 2023.

---

### Meta-Review · Area_Chair_qGEg · 2024-12-21

**Metareview:**

This paper uses a Time-series Patch Compression (TPC) module to decompose time series patches into mode functions, capturing representations of long-and-short-term features. Next, a diffusion model with a CNN backbone is models changes to the latent distributions and is used to generate multivariate long-term time series. The model is tested on standard benchmarks in time-series modeling and generation. The primary claim for this work is that CPDD "addresses the challenges of balancing long-term temporal dependencies and short-term feature representations by integrating the TPC module with a diffusion-based generative model". The integration of diffusion based models with existing

Overall, this was an empirically driven deep learning paper and at the end of the reviews and discussion period remained a borderline paper that fell on the side of rejection. Despite the commendable changes the authors made to the manuscript (which have improved it), I think it came down to a lack of clarity on precisely how the TPC module helps balance long vs short term features. I think this can potentially be improved by a rewrite of the manuscript to highlight this aspect as well as thorough experimentation along the ablations recommended by reviewer fwsP.

**Additional Comments On Reviewer Discussion:**

The reviewers requested comparisons of computational complexity between CPDD and diffusion models to understand if CPDD reduces training and inference time despite incorporating a Transformer. The authors provided tables comparing training and sampling times on multiple datasets, showing shorter runtimes. The reviewers requested more standard metrics (e.g., MSE, MAE) and further comparisons with additional diffusion- and Transformer-based baselines. The authors agreed that standard metrics alone are insufficient for generative tasks and included Context-FID and Correlational scores. The overall CPDD pipeline was unclear, especially regarding the TPC module’s reconstruction loss, and whether CPDD is trained end-to-end or in two stages.They provided pseudocode for both stages and reaffirmed that the TPC loss function will be explicitly added to the final manuscript.

Perhaps the most important point the reviewers raised was that they were not able to come to a satisfactory conclusion on was this idea of “balancing” long-term dependencies with short-term feature learning. Specifically, it was unclear how CPDD distinguished itself from existing multi-scale methods. The response stated that DSConv handled local (intra-patch) patterns, while the Transformer captures global (inter-patch) dependencies. The core claim by the authors is that unlike typical multi-scale methods, CPDD specifically addresses optimization challenges for generative modeling by decomposing the learning objectives. However, I think this last point remained unclear from my reading of the revised manuscript.

---

### Decision · Program_Chairs · 2025-01-22

Reject